

# Potentially inappropriate medication and its associated factors in older people living in nursing homes: a cross-sectional study

Ester Goutan-Roura[1,2,*], Geovanna O. Carneiro[3,*], Francisca S.M. Moreira[3], Montse Masó-Aguado[1,2], Pau Moreno-Martin[1,2], Eduard Minobes-Molina[1,2], Dawn A. Skelton[4] and Javier Jerez-Roig[1,2]

[1] Institute for Research and Innovation in Life Sciences and Health in Central Catalonia (IRIS-CC), Vic, Spain
[2] Research Group on Methodology, Methods, Models and Outcomes of Health and Social Sciences (M₃O), Faculty of Health Sciences and Welfare, Centre for Health and Social Care Research (CESS), University of Vic-Central University of Catalonia (UVic-UCC), Vic, Spain
[3] Department of Pharmaceutical Sciences, Federal University of Pernambuco, Recife, Brazil
[4] Research Centre for Health (ReaCH), School of Health and Life Sciences, Glasgow Caledonian University, Glasgow, United Kingdom
* These authors contributed equally to this work.

Corresponding author
Ester Goutan-Roura,
ester.goutan@uvic.cat

## ABSTRACT

**Background.** Institutionalized residents tend to use more drugs and in larger doses. Potentially inappropriate medications (PIM) use is highly prevalent among them. In addition, they are more likely to be prescribed multiple medications (polypharmacy). Moreover, many drugs considered PIM have increased anticholinergic burden (ACB), responsible for adverse drug events (ADE).

**Objective.** Identifying PIM, polypharmacy and ACB among older people's prescriptions as well as their associated factors.

**Methods.** Cross-sectional observational multicentre study. Drug information was collected from the nursing homes, medical registers. PIM exposure was assessed using Beers Criteria 2023. ACB was calculated using the Anticholinergic Risk Scale. Other sociodemographic, as well as health-related data were also collected.

**Results.** 130 residents (83.8% women) mean age 85.1 (±7.4). Over 80% (111) of residents have prescriptions including at least one PIM. Polypharmacy (≥5 drugs) occurred in 69.1% (94), while extensive polypharmacy (≥10 drugs) occurred in 18.4% (25). The most prevalent PIMs were benzodiazepines (57.3%; 73), antipsychotics (48.5%; 66) and proton pump inhibitors (39.7%; 54). Regarding ACB, 63.1% (82) of the residents have prescriptions including at least one anticholinergic drug. In the multivariate analysis, ACB ($p = 0.018$; OR 3.52) and polypharmacy (p=0.015; OR 3.58) were associated with PIM.

**Conclusions.** The prevalence of PIM, polypharmacy and ACB was very high (84%, 69%, and 63% respectively) in this sample of nursing home residents. ACB and polypharmacy were significantly associated with PIM. Anticholinergic drugs should be carefully assessed and gradually withdrawn when not needed. Balancing treatment with other biopsychosocial interventions may contribute to reducing polypharmacy.

## INTRODUCTION

According to the World Health Organization (WHO), between 2020 and 2050 the world's population of people aged 60 years and older will double (to 2.1 billion). In the same period, the number of people aged 80 years or older is expected to triple to 426 million (*World Health Organization, 2022*). This population ageing is associated with chronic health problems, incapacity, multimorbidity and greater investments in health resources (*Garcia, Medina-Bustos & Schiaffino, 2016*), including nursing homes.

Institutionalized residents tend to use more drugs and in larger doses, and they usually show other characteristics (poor mental health, frailty, or lower physical activity, among others) which exposes them to drug-related problems (*Iniesta-Navalón et al., 2019*). The term "potentially inappropriate medications (PIM)" refers to those drugs which should not be prescribed in certain conditions for specific populations because the risk of adverse events outweighs the clinical benefit (*Iniesta-Navalón et al., 2019*; *Laroche, Charmes & Merle, 2007*). Different prevalence of PIM has been described depending on the health setting, identification tool used, or country (*Laroche, Charmes & Merle, 2007*). According to *Bories et al. (2021)*, PIM prevalence in a hospital setting was higher than that in nursing homes and in primary care (44%, 29% and 19% respectively) (*Storms et al., 2017*).

PIM use is associated with negative health outcomes such as adverse drug events (ADE), falls, cognitive impairment, hospitalization, and mortality (*Lohman et al., 2017*). Regarding hospitalization, *Lohman et al. (2017)* showed that patients using at least one PIM had a 13% greater risk of being hospitalized than patients using no PIMs, while patients using at least two PIMs had a 21% greater risk. Moreover, the higher the dose of PIM being prescribed, the higher the risk for drug-drug interactions (DDI) (*Forgerini et al., 2021*). DDIs can compromise the patient's health and lead to emergency department admission and hospitalization (*Becker et al., 2001*).

Older adults are more likely than their younger counterparts to be prescribed multiple medications due to age-related multimorbidity, frailty, and other geriatric syndromes (*Mohile et al., 2011*; *Wilkinson & Izmeth, 2023*; *Romskaug et al., 2020*). Polypharmacy consists of five or more drugs intake and its prevalence is estimated at around 50% in the aged population. Differences can be found depending on different definitions of polypharmacy, extensive polypharmacy (>10 drugs) and health study settings. Polypharmacy is associated with PIM, DDI, longer length of hospital stays and increased hospital readmission rate and mortality (*Almeida et al., 2019*; *Piccoliori et al., 2021*; *Steinman et al., 2006*; *Hilmer & Gnjidic, 2009*; *Nobili, Garattini & Mannucci, 2011*).

As described in *Novaes et al. (2017)* older populations are at higher risk for presenting the "iatrogenic triad": polypharmacy, PIM and DDI. Each of these conditions leads to poor health outcomes and increased healthcare costs. Moreover, many drugs considered PIMs present an increased anticholinergic burden (ACB) and are responsible for ADE such as blurred vision, dry mouth, falls, and cognitive and physical decline (*Rudolph et al.,*

*2008*). According to *Bourrel et al. (2020)*, nearly half of the residents in their study having a prescription with a high ACB were taking more than nine drugs and all the prescriptions with more than five PIMs had anticholinergic burden, with the majority having a high anticholinergic burden. Healthcare providers must guarantee a safe medication intake and risk reduction of ADE for the ageing population. Identifying PIM, polypharmacy and ACB among older people's prescriptions as well as their associated factors is the first step to guarantee a safer drug intake.

As far as we know, there is no existing literature analyzing the use of PIMs by American Geriatrics Society (AGS) Beers 2023 criteria and anticholinergic load in nursing homes. The study aimed to describe Catalan nursing homes, population characteristics, their PIMs and PIM-associated factors, polypharmacy and ACB in their prescriptions, so that better clinical decisions could be made.

## MATERIALS & METHODS

### Study design and population

This is a cross-sectional observational multicentre study using data from the OsoNaH project, registered in Clinical Trials through the ID number NCT04297904 (date of first registration: 06/03/2020); further details are published elsewhere (*Farrés-Godayol et al., 2021*). The "Strengthening the Reporting of Observational studies in Epidemiology" (STROBE) guidelines were adhered to as in *Von Elm et al. (2007)*. The study was approved by the University of Vic-Central University of Catalonia (UVic-UCC) Ethics and Research Committee (92/2019) and Clinical Research Ethics Committee of the Osona Foundation for health research and education (FORES) (code 2020118/PR249). The study was conducted according to the Declaration of Helsinki, as well as the Organic Law 3/2018 (5 December) on the Protection of Personal Data and Guarantee of Digital Rights in Spain.

At the start of the study, 19 nursing homes were registered in Osona County. Of these, six declined to participate, while the rest agreed to take part.

However, due to COVID-19 restrictions, data collection had to be discontinued in March 2020, resulting in a final dataset comprising five nursing homes: three subsidized and two for-profit facilities.

All residents aged 65+ permanently living in the nursing homes, consenting, and signing the informed consent were included. For those with cognitive incapacity, consent was received from the legal guardian. Subjects in a coma or palliative care (short-term prognosis), those hospitalized during data collection (conducted during January-March 2020) and those lacking medication data were excluded from the study. A minimum of 110 individuals were necessary, according to a prevalence of PIMs of 92.3% found in a study also conducted in Catalonia (*Molist-Brunet et al., 2021*; *World Health Organization, 2012*). We considered a 95% confidence level and 5% margin of error for this calculation.

The nursing homes, records were used to collect information related to drugs. The Anatomical Therapeutic Chemical (ATC) classification system was used to classify the active ingredients of drugs according to the organ/system on which they act and their therapeutic, pharmacological, and chemical properties (*World Health Organization, 2012*).

The dependent variable in the study was the use of at least one PIM by older adults.

### PIM exposure

PIM use was assessed using Beers Criteria—"2023 American Geriatric Society Beers Criteria® Potentially Inappropriate Medication Use in Older Adults"—which includes medications to be avoided in most older adults, regardless of underlying diseases (*AGS, 2023*). As the dosing and indication parameters weren't available, insulin, aspirin, and digoxin were not included in the PIM analysis.

### Polypharmacy

Polypharmacy (five or more drugs) and extensive polypharmacy (>10 drugs) were identified from nursing homes, medical registers.

### Anticholinergic activity

Each participant's anticholinergic burden (ACB) was calculated using the Anticholinergic Risk Scale (ARS), which estimates the extent to which an individual patient is exposed to a higher risk of anticholinergic adverse effects that has a negative impact on the elderly population (*Rudolph et al., 2008*). Weightings for each medication (0–3 points) were taken from the published scale description and then summed to provide a numeric value for each participant. Being (0) limited or no anticholinergic potential, (1) moderate, (2) high and (3) very high.

### Covariates

Sociodemographic and health-related information were obtained from the nursing homes, registers and checked with the nursing homes, professionals, including age, sex, education level, marital status, type of nursing homes (private or subsidized), number of months institutionalized, smoking and drinking habits, body mass index (BMI), weight loss during the last 12 months, chronic conditions, retrospective hospitalizations and fractures in the last 12 months, urinary tract infections in the last 30 days, ulcers, and delirium episodes in the last year. Chronic conditions were high blood pressure, diabetes, cancer, lung disease, stroke, dementia, Parkinson, osteoporosis, kidney failure, heart disease, mental illness, depression, dyslipidaemia, hypothyroidism, hyperthyroidism, anxiety, vertigo, chronic pain, hypotension, visual impairment, epilepsy, fibromyalgia, hyperparathyroidism, vaginal prolapse, prostate hyperplasia, anaemia, osteoarthritis, sleep disorders, hernia hiatus, lupus, sarcopenia, hearing loss, circulatory disease, digestive disease.

Cognitive status was assessed using the Spanish-validated Pfeiffer Scale (*De La Iglesiaa et al., 2001*). Nutritional status was assessed using the Mini Nutritional Assessment (MNA) test, a validated tool which classifies individuals into three groups: 'normal nutritional status' (≥24), 'at risk of malnutrition' (17–23.5) and 'malnourished' (<17) (*Guigoz, Vellas & Garry, 1997*).

Physical performance was assessed with the Short Performance Physical Battery (SPPB, an objective assessment tool for evaluating lower extremity functioning in older persons) (*Guralnik et al., 1994*); the Clinical Frailty Scale (CFS, a judgement-based frailty tool that assesses the person's illnesses, function and cognition to generate a frailty score ranging from 1 (very fit) to 9 (terminally ill)), was used to assess frailty severity, with a focus on functional capacity and daily living activities. It helps identify patients at increased risk of

adverse outcomes, enabling timely preventive interventions to optimise care (*Rockwood et al., 2005*).

## Statistical analysis

Descriptive analysis was conducted providing absolute and relative frequencies, as well as mean and standard deviation (SD) for categorical and quantitative variables, respectively. The Chi-square was used for bivariate analysis, and the variables with a significance value lower than 0.20 were tested for the multivariate analysis. The latter was carried out using binary logistic regression to detect the factors associated with PIMs utilization (as a dichotomous variable), following the forward method. The variable was retained in the final model if it maintained statistical significance ($p < 0.05$) or acted as a confounder by adjusting a covariate, such as reducing the odds ratio (OR) of a significant variable, in the multivariate analysis. The goodness-of-fit was examined with the Hosmer-Lemeshow test, such that $p > 0.05$ indicated a proper adjustment of the model. All the statistical analysis was conducted with the IBM SPSS Statistics software (IBM Corp. Released 2021. IBM SPSS Statistics for Windows, Version 28.0. IBM Corp.: Armonk, NY, USA).

## Ethical approval and consent to participate

Ethics and Research Committee of the University of Vic - Central University of Catalonia gave Ethical permission (registration number 92/2019). Residents or his/her legal guard signed the informed consent. Relevant guidelines and regulations were followed in all methods performed.

## RESULTS

From the 185 potential residents, 55 (29.4%) did not meet the inclusion criteria for the study (Fig. 1. Flowchart of the sampling process of nursing home residents (Osona, Spain, 2020)).

Among 130 residents, the mean age was 85.1 ($\pm$ 7.4) years, mostly female 83.8%, with 38.0 ($\pm$44.9) institutionalised months. All of the residents in the analysis were diagnosed with at least one chronic condition, with a mean of 5.1 $\pm$2.4 diagnosed comorbidities. The most prevalent medical conditions were arterial hypertension (63.6%; 83), dementia (55.0%), dyslipidaemia (31.0%), diabetes (27.9%) and depression (27.9%). Table 1 presents data on the sociodemographic characteristics, clinical features, and medical conditions of the residents.

The clinical assessment of cognitive and functional capacity reveals that 79.4% of the residents presented cognitive decline and decreased functional capacity and frailty. Sarcopenia was found in 76.0% of the residents. In the last 12 months, 45.0% of the residents have fallen at least once, 37.1% had episodes of delirium, 18.9% had lost weight, 18.9% had skin lesions, 19.4% were hospitalized, 8.7% had a bone fracture and 4.5% had leg ulcers.

Polypharmacy ($\geq$5 drugs) occurred in 69.1% of the residents with the mean number of drugs being 13.1 $\pm$ 6.2, while extensive polypharmacy ($\geq$10 drugs) occurred in 18.4% of the residents. Over 80% of the residents have prescriptions including at least one PIM. Relying

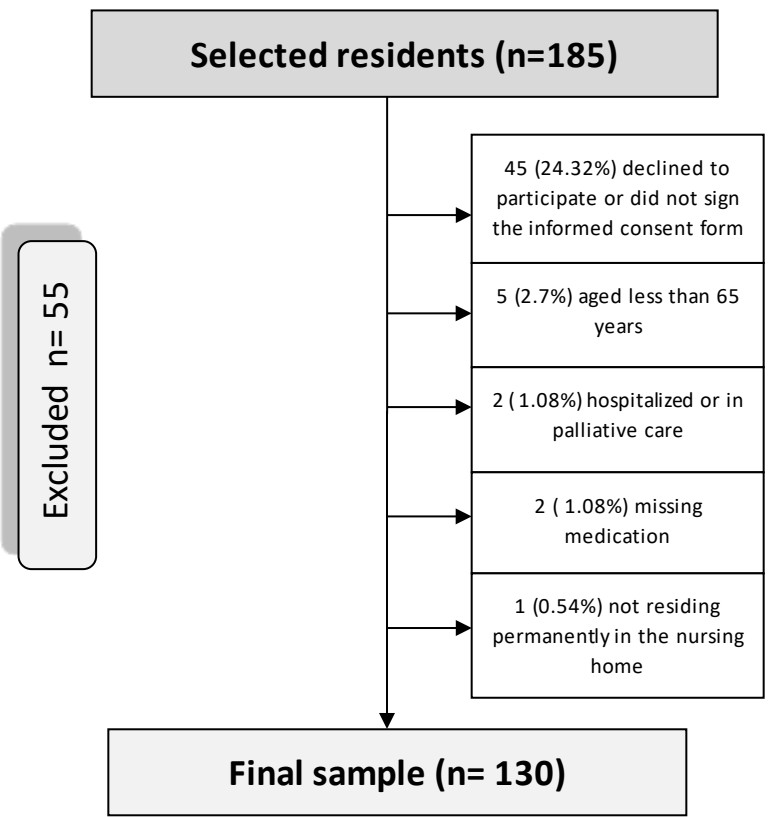

**Figure 1** Flow chart of the sampling process of nursing home residents.

on AGS Beers Criteria (2023) a total of 23 PIMs were identified. The most prevalent PIMs were related to drugs that act on the central nervous system (CNS), benzodiazepines were the most detected PIM class (57.3%), followed by antipsychotics (48.5%) and proton pump inhibitors (PPI) (according to utilization longer than 8 weeks criteria; 39.7%). In respect of ACB, 63.1% of the residents have prescriptions including at least one anticholinergic drug. The most prevalent PIM according to AGS Beers 2023 and the drugs' ACB are reported in Table 2.

Table 3 shows the results of the bivariate analysis between PIM and independent variables with $p$-value lower than 0.20. Moderate-high ACB and polypharmacy were the variables significantly associated ($p < 0.05$) with PIM in the bivariate analysis and their statistical significance remained in the multivariate analysis. The value of the Hosmer-Lemeshow test was 0.46, indicating proper adjustment of the final model.

## DISCUSSION

We aimed to describe PIMs use and its associated factors among older people living in nursing homes from Central Catalonia (Spain). This study reveals an alarmingly high prevalence of PIM use among older adults in nursing homes. In fact, eight out of 10 residents present at least one PIM—benzodiazepines, antipsychotics, and PPIs are the most

**Table 1  Socio-demographic and health-related information of older people living in five Catalan nursing homes.**

|  | *n* | % |
|---|---|---|
| **Educational level** | | |
| Illiterate | 38 | 29.5 |
| Primary school | 59 | 38.8 |
| High school | 7 | 5.4 |
| College education | 4 | 3.3 |
| Unknown | 31 | 23.3 |
| **Marital status** | | |
| Single | 17 | 13.1 |
| Married/dating | 19 | 14.6 |
| Divorced | 3 | 2.3 |
| Widow(er) | 76 | 58.5 |
| Unknown | 16 | 11.5 |
| **Body mass index** | | |
| Underweight | 26 | 28.6 |
| Normal/overweight | 45 | 48.4 |
| Obese | 21 | 23.1 |
| **Frailty** | | |
| Very well | 1 | 0.8 |
| Well | 15 | 11.5 |
| Managing well | 5 | 3.8 |
| Vulnerable | 6 | 4.6 |
| Mily frail | 22 | 16.9 |
| Moderately frail | 39 | 29.2 |
| Severely frail | 33 | 25.4 |
| Very severely frail | 10 | 7.7 |
| **SPPB**[*] | | |
| Robustness | 5 | 4.1 |
| Pre-frailty | 18 | 14.6 |
| Frailty | 29 | 23.6 |
| Disability | 71 | 58.7 |
| **Cognitive state** | | |
| Intact | 26 | 20.6 |
| Slight cognitive impairment | 15 | 11.9 |
| Moderate cognitive impairment | 29 | 23.0 |
| Severe cognitive impairment | 56 | 44.4 |
| **Nutritional state** | | |
| Normal | 30 | 27.8 |
| At risk of malnutrition | 69 | 63.9 |
| Malnourished | 9 | 8.3 |
| **Comorbidities** | | |
| Arterial hypertension | 83 | 63.3 |
| Dementia | 71 | 55.0 |

**Table 1** (*continued*)

|  | *n* | % |
|---|---|---|
| Dyslipidaemia | 41 | 31.0 |
| Diabetes | 36 | 27.9 |
| Depression | 36 | 27.9 |
| Stroke | 26 | 20.2 |
| Cancer | 25 | 19.4 |
| Lung disease | 24 | 18.6 |
| Osteoporosis | 22 | 17.1 |
| Digestive disease | 22 | 17.1 |
| Mental disease | 21 | 16.3 |
| Parkinson's disease | 18 | 14.0 |
| Arthrosis | 18 | 14.0 |
| Hypothyroidism | 18 | 14.0 |
| Anaemia | 17 | 13.2 |
| Circulatory disease | 16 | 12.4 |
| Anxiety | 9 | 7.0 |
| Chronic pain | 9 | 7.0 |
| Sleep disorders | 6 | 4.7 |

**Notes.**
*SPPB: Short Performance Physical Battery.

recurring-. This is coincident with other studies which report a PIM prevalence range of 54.6% to 90.6% (*Moreira et al., 2020*; *Jankyova, Rubintova & Foltanova, 2020*).

Our PIM analysis is based on Beers Criteria 2023, which is the revision of Beers Criteria 2019 by an interprofessional expert panel including the published evidence update, new criteria and modification of existing ones together with formatting changes to enhance usability. AGS Beers criteria apply to 65 years and older and in institutionalized settings of care, which is where our studied population lives (*AGS, 2023*). Aged people, with multiple diseases and taking multiple medications are among the main reasons for this high PIM prevalence found in Catalan nursing homes.

In our study, benzodiazepines, antipsychotics, and PPIs were the most prevalent PIMs. This is consistent with findings reported in the literature. A study conducted in Malaysia by *Azri et al. (2016)* reported that more than 70% of institutionalized older adults had poor sleep quality and were prescribed benzodiazepines, despite the well-documented worrying adverse effects for this population and being the PIM most used worldwide for insomnia and anxiety. Antipsychotic drugs are commonly prescribed to manage poor sleep and behavioural and psychological symptoms associated with dementia; this occurs despite their association with falls and stroke and increased mortality risk (*Lucchetti & Lucchetti, 2017*; *Westbury et al., 2018*; *Sterke et al., 2012*; *Schneeweiss et al., 2007*). A high prevalence of antipsychotics was also reported by *Moreira et al. (2020)* as the most prevalent PIM considering the Beers Criteria 2015.

Another reason for concern is the high prevalence of proton pump inhibitors (PPI) use, as well as its association with certain diseases (*Nishtala & Soo, 2015*; *Yu et al., 2017*; *Lai et al., 2019*). Despite the high prevalence of PPI, according to a critical review, 25% to

**Table 2** Most prevalent Potentially Inappropriate Medications (PIMs) and Anticholinergic Burden: A Dual Analysis Using AGS Beers 2023 and the Anticholinergic Risk Scale (ARS).

| Medication | N (%) | PIM Status (Beers Criteria - *Rationale*) | Anticholinergic activity (ARS) | Overlap |
|---|---|---|---|---|
| Benzodiazepine | 73 (56.2) | Yes | – | No |
| *Lorazepam* | *47 (36.2)* | Older adults have increased sensitivity to benzodiazepines | | |
| *Lormetazepam* | *11 (8.5)* | and decreased metabolism of long-acting agents; | | |
| *Alprazolam* | *7 (5.4)* | the continued use of benzodiazepines may lead to | | |
| *Diazepam* | *3 (2.3)* | clinically significant physical dependence. In general, all | | |
| *Clorazepato* | *2 (1.5)* | benzodiazepines increase the risk of cognitive impairment, | | |
| *Midazolam* | *2 (1.5)* | delirium, falls, fractures, and motor vehicle crashes in older | | |
| *Clonazepam* | *1 (0.8)* | adults | | |
| | | | | |
| Antipsychotics | 66 (48.5) | Yes | | Yes |
| *Quetiapine* | *29 (22.3)* | Increased risk of cerebrovascular accident (stroke) and | 1 | |
| *Risperidone* | *24 (18.5)* | greater rate of cognitive decline and mortality in persons | 1 | |
| *Olanzapine* | *7 (5.4)* | with dementia. Additional evidence suggests an association | 2 | |
| *Haloperidol* | *3 (2.3)* | of increased risk between antipsychotic medication and | 1 | |
| *Aripiprazole* | *1 (0.8)* | mortality independent of dementia. | – | |
| | | | | |
| Proton pump inhibitors | 54 (41.5) | Yes | – | No |
| *Omeprazole* | *48 (36.9)* | Risk of *Clostridioides difficile* infection - pneumonia, | | |
| *Lansoprazole* | *4 (3.1)* | gastrointestinal malignancies, - bone loss, and fractures. | | |
| *Pantoprazole* | *1 (1.5)* | | | |
| | | | | |
| Antidepressants | 28 (21.5) | [*]Only Paroxetine and amitriptyline: | | No |
| *Trazodona* | *15 (11.5)* | Highly anticholinergic, sedating, and cause orthostatic | 1 | |
| *Mirtazipine* | *9 (6.9)* | hypotension; | 1 | |
| *Paroxetine*[*] | *3 (2.3)* | | 1 | |
| *Amitriptyline*[*] | *1 (0.8)* | | 3 | |

**Notes.**
[*]Beers criteria rationale is only referred to [*] drugs: paroxetine and amitriptyline.

75% of these prescriptions have no indication, implying that its utilization should be more critically evaluated and maybe stopped (*Batchelor et al., 2017*). Coincident with the present study results, benzodiazepine and PPI were also found among the most prevalent PIMs in the data published by *Almeida et al. (2019)*. A higher prevalence of PIM use is positively associated with drugs with a high anticholinergic burden and polypharmacy (*Moreira et al., 2020*).

In our study, polypharmacy showed statistical significance in the multivariate analysis. A balance is required between over- and under-prescribing to achieve optimal outcomes in challenging situations, such as the management of chronic conditions in older people. In other published studies, polypharmacy represents one of the most frequently associated factors with PIM use (*Komiya et al., 2018*). Therefore, it is particularly important to reconsider medication appropriateness in late life, but the medication service review approach is not well implemented in all nursing homes in Catalonia (*Spinewine et al., 2007*).

Our multivariate analysis indicates that moderate-high anticholinergic activity was significantly associated with PIM. Anticholinergic drugs are considered the first-place

**Table 3** Bivariate and multivariate analysis between potentially inappropriate medications and independent variables among older people living in nursing homes.

| Variable | n | % | p* | Odds Ratio (Confidence Interval 95%)* | p (adj.)** | Odds Ratio (adj.) (Confidence Interval 95%)** |
|---|---|---|---|---|---|---|
| **Polypharmacy** | | | | | | |
| *Yes* | 94 | 72.3 | 0.012 | 3.63 (1.33–9.89) | 0.015 | 3.58 (1.27–10.07) |
| *No* | 36 | 27.7 | | | | |
| **Anticholinergic activity** | | | | | | |
| *Moderate or high* | 82 | 63.1 | 0.014 | 3.57 (1.30–9.83) | 0.018 | 3.52 (1.24–9.98) |
| *No anticholinergic burden* | 48 | 36.9 | | | | |
| **Depression** | | | | | | |
| *Yes* | 36 | 27.7 | 0.085 | 3.80 (0.83–17.38) | | |
| *No* | 94 | 72.3 | | | | |
| **Parkinson** | | | | | | |
| *Yes* | 18 | 14.0 | 0.076 | 0.34 (0.10–1.12) | | |
| *No* | 111 | 86.0 | | | | |
| **Delirium** | | | | | | |
| *Yes* | 47 | 36.2 | 0.147 | 2.37 (0.74–7.62) | | |
| *No* | 83 | 63.8 | | | | |

**Notes.**
*Bivariate analysis (only variables with p-value below 0.20 are shown).
**Multivariate analysis.

risk factor for drug-related problems in older people (*Collamati et al., 2015*). Normal age-related declines in memory could increase susceptibility to the potential cognitive side effects of drugs with high anticholinergic activity, and its consumption is associated with frailty, deterioration in quality of life and an increase in morbidity and mortality (*Spinewine et al., 2007*; *Collamati et al., 2015*; *Murali Doraiswamy et al., 2002*). Comorbid conditions common in older patients, including Parkinson's disease, Alzheimer's dementia, delirium, may also make these residents more susceptible to cognitive impairment and exaggerate the effects of anticholinergic drugs on cognitive function (*Collamati et al., 2015*). There is no easy way to assess the anticholinergic burden to advise clinical practice. Diverse tools have been developed to estimate the cumulative effects of drugs with anticholinergic effects based on either expert consensus, serum anticholinergic activity or pharmacological principles (*Hilmer & Gnjidic, 2022*). In other reports, high ARS scores were associated with the lower score for various components of the Barthel index (*Lowry et al., 2011*).

In our analysis, approximately six out of 10 participants were exposed to some anticholinergic activity, and antipsychotics were the majority of them (according to ARS). AGS Beers Criteria mentions drugs with high anticholinergic properties but does not specify the burden. For this reason, this work opted to use ARS. More than half of the residents (63.1%) included in this study have a moderate or high ACB with potential clinical implications. Worth noting that among those that have prescriptions of drugs with moderate or high ACB, 92.7% have PIM prescription (since many of the drugs that are PIM also have anticholinergic activity—in our study antipsychotics and

antidepressants) and 74.4% use five or more drugs per day. Quantifying the extent to which PIMs and anticholinergic burden overlap in this population (Table 2) aims to determine whether deprescribing PIMs—a modifiable risk factor—could simultaneously reduce both inappropriate prescribing and anticholinergic burden, thereby prioritizing interventions with dual clinical benefits.

The WHO highlights that nursing homes should prioritise maintaining the functional capacity of residents to support healthy ageing. This involves enhancing or compensating for declines in intrinsic capacity through appropriate support and environmental care (*World Health Organization, 2015*). Beyond meeting basic survival needs, nursing homes should implement multicomponent interventions, including physical, cognitive, nutritional, and pharmacological strategies, to promote residents' well-being. Regarding the prescription of medicines, clinical indications to prescribe anticholinergic drugs should be carefully examined, and a slow and gradual withdrawal (carefully monitored) should be started when indications are no longer present (*Bourrel et al., 2020*; *Andrew et al., 2018*).

Beers criteria, together with ACB of drugs, are important tools to assist decision-making in assessing geriatric patients' prescriptions. The findings of this study highlight the need for nursing homes to operationalize a medication service review targeting PIMs, polypharmacy and ACB drugs, particularly antipsychotics and antidepressants.

### Strengths and limitations

To our knowledge, this study was the first to analyse PIM use by AGS Beers Criteria 2023, polypharmacy and anticholinergic burden with ARS in Catalan nursing homes for older adults. We found that polypharmacy and anticholinergic medication are associated factors to PIM, explained by the need to treat the many comorbidities present in this population.

As a limitation, it should be recognized that insulin, aspirin, and digoxin were not considered for PIM analysis, since dose and indication were not collected. This could have led to the sub-estimation of PIM. Although the sample size included in this study was initially enough to verify the prevalence of PIMs, it may be insufficient to explore all of the associated factors. In this sense, the restrictions caused by the covid-19 pandemic prevented us from continuing including more nursing homes in the project.

## CONCLUSIONS

In nursing homes, it is crucial to carefully review PIMs, polypharmacy, and thoroughly assess the need for anticholinergic drugs as these medications can increase the risk of cognitive decline, falls, and other adverse effects, particularly in the elderly population. Effective interdisciplinary collaboration between healthcare providers, regular medication reviews and deprescribing practices should be prioritized to reduce the burden of polypharmacy in nursing homes. When no longer necessary, a slow and monitored withdrawal should be initiated to ensure patient safety. It is also important to balance pharmacological treatment with other biopsychosocial interventions, such as cognitive therapy, physical activity programs, and social engagement, to provide comprehensive and integral care for nursing home residents.

## ACKNOWLEDGEMENTS

We thank all the participating nursing homes for their contribution to this work and the older adults who participated in the study.

### Funding

This work was funded by Hestia Chair from Universitat Internacional de Catalunya (grant number BI-CHAISS-2019/003) and the Catalan Board of Phisiotherapists (Code: R03/19). The funders had no role in study design, data collection and analysis, decision to publish, or preparation of the manuscript.

### Grant Disclosures

The following grant information was disclosed by the authors:
Hestia Chair from Universitat Internacional de Catalunya: BI-CHAISS-2019/003.
The Catalan Board of Phisiotherapists: Code: R03/19.

### Competing Interests

The authors declare there are no competing interests.

### Author Contributions

- Ester Goutan-Roura conceived and designed the experiments, performed the experiments, analyzed the data, prepared figures and/or tables, authored or reviewed drafts of the article, and approved the final draft.
- Geovanna O. Carneiro conceived and designed the experiments, analyzed the data, prepared figures and/or tables, authored or reviewed drafts of the article, and approved the final draft.
- Francisca S.M. Moreira conceived and designed the experiments, prepared figures and/or tables, authored or reviewed drafts of the article, and approved the final draft.
- Montse Masó-Aguado conceived and designed the experiments, performed the experiments, authored or reviewed drafts of the article, and approved the final draft.
- Pau Moreno-Martin conceived and designed the experiments, performed the experiments, analyzed the data, authored or reviewed drafts of the article, and approved the final draft.
- Eduard Minobes-Molina conceived and designed the experiments, performed the experiments, analyzed the data, authored or reviewed drafts of the article, and approved the final draft.
- Dawn A. Skelton conceived and designed the experiments, authored or reviewed drafts of the article, and approved the final draft.
- Javier Jerez-Roig conceived and designed the experiments, performed the experiments, analyzed the data, prepared figures and/or tables, authored or reviewed drafts of the article, and approved the final draft.

## Human Ethics

The following information was supplied relating to ethical approvals (i.e., approving body and any reference numbers):

The Ethics and Research Committee of the University of Vic - Central University of Catalonia gave Ethical permission (registration number 92/2019). Residents or his/her legal guard signed the informed consent.

## Data Availability

The raw data is available in the Supplemental File.

## Supplemental Information

Supplemental information for this article can be found online at http://dx.doi.org/10.7717/peerj.19570#supplemental-information.

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
