# Peer review of "Potentially inappropriate medication and its associated factors in older people living in nursing homes: a cross-sectional study"

_PeerJ, doi:10.7717/peerj.19570_

## Round 0.1 · original submission · Minor Revisions

The authors should address the comments of Reviewer 2, and make sure that each of the potential areas of confusion are clarified. In addition, the authors could probably take a slightly more critical view of some of the literature they cite - for example, the papers about PPI and Parkinsons Disease/dementia seems to show small effects in limited populations with at times poorly controlled studies (with no effects in larger populations). I also note that neither levodopa nor selegiline are on the Beers list, so why/were they included in this analysis. Finally, the authors comment that drugs were being used without an indication ... it is would be useful to comment to what degree the information analysed was likely to be complete - do they have any feeling about how complete the data they analyzed was ? Incomplete information would be a potential limitation of the study.

Reviewer 1 ·

Basic reporting

Thank you so much for giving this opportunity to review this amazing paper. I thank the authors for doing such great work. I think this work is great with some minor modifications. PIM is a crucial health phenomena among older adult population and should have been taken seriously by health care providers. Doing such a study of it's first in a kind, would definitely add to the literature. Overall, the text needs minor english editing, the literature is sufficient and the flow is logical.

Experimental design

The research question is valid and original and worth studying. The methods were described in a detailed manner. Although the authors mentioned that the researcher collected data on a very long list of medication conditions, no report or table were utilized in the results sections? I would suggest to either calculate CCI or present the prevalence of these condition in a separate table.

Validity of the findings

Although the results were written in detail, not that much text were written on table-3, which is very important table. Rethink of including more text on table-3 and again talk about the other medical conditions you mentioned in the methods. Overall, the results section is clear and reflect what the researcher have found.

Additional comments

Good luck.

Reviewer 2 ·

Basic reporting

(1) Abstract: method and results
The authors should describe that you performed logistic regression analysis in the methods section of the abstract. They also describe that polypharmacy is a related factor of PIMs in the results, but what is the basis for this description? If the authors set the statistical significance level at 0.05, it is not significant.

(2) Materials and Methods: Study design and population (Line99-101)
What kind of facilities were the five facilities included in this study? It would be desirable to add a description of the characteristics and representativeness of these facilities.

(3) Figure1, Materials and Methods: Study design and population (Line111-112)
The exclusion criteria in the Methods section do not seem to match the explanation in Figure 1. It is recommended that you provide sufficient and necessary descriptions in the main text as well.

(4) Discussion: Strengths and limitations (Line 312-313)
It is described that ‘...aspirin and digoxin were not considered for PIMs analysis since dose and indication were not collected’, but in the methods section, insulin is listed instead of aspirin. This should be checked to see which is the correct.

(5) Table 1
Regarding Table 1, the authors could make it easier for readers to understand the contents by also including a column dividing the data into PIM and non-PIM.

(6) Table 3
The bivariate analysis in Table 3 includes a variable with a P value of 0.202. What criteria were used to select this variable? Please clarify this point.
Minor comments

(7) It is preferable to spell out NHs in the main text as well as in the abstract.

(8) It is preferable to spell out abbreviations such as OR and CI in Table 3.

Experimental design

(1) Materials and Methods: Statistical analysis (Line 173-177)
The intention to conduct multivariable analysis was unclear. In examining the relationship between exposure (resident characteristics) and outcome, I think it is necessary to calculate estimates after adjusting for any possible confounding factors. It is unclear to what extent the process used in this study takes into account confounding factors. There are also some uncertainties regarding the final model selection. According to the description in the main text, it seems that the final model was selected based on the Hosmer-Lemeshow test, but was this determined by referring to the P-value? Since the P-value is not thought to indicate the degree of goodness of fit, it may consider using information measures such as AIC or BIC. In any case, more detailed descriptions of the analysis model are required.
Personally, I think the results of univariate analysis are interesting enough even without multivariable analysis.

Validity of the findings

none

Additional comments

(1) Overall
The relationship between PIMs and ACB is discussed, but since PIMs include anticholinergic drugs, it seems natural that there is a high proportion of high ACB in the PIMs group. The authors should clarify the purpose of their exploration of the relationship between PIMs and ACBs.

Reviewer 3 ·

Basic reporting

No comment

Experimental design

No comment

Validity of the findings

No comment

Additional comments

I commend the authors for their well designed and new insights regarding the PIMs use among elderly. The manuscript was clearly written in professional, unambiguous language.

I had provided a few comments in the shared PDF to consider, which I believe should be addressed before acceptance.

Annotated reviews are not available for download in order to protect the identity of reviewers who chose to remain anonymous.

---

## Round 0.2 · accepted · Accept

I am happy that the authors have appropriately addressed the comments of reviewers. The language has also been clarified. The paper is ready to go.